# Comparative Analysis and Structural Modeling of *Elaeis oleifera* FAD2, a Fatty Acid Desaturase Involved in Unsaturated Fatty Acid Composition of American Oil Palm

**DOI:** 10.3390/biology11040529

**Published:** 2022-03-30

**Authors:** Rayda Ben Ayed, Tejas Chirmade, Mohsen Hanana, Khalil Khamassi, Sezai Ercisli, Ravish Choudhary, Narendra Kadoo, Rohini Karunakaran

**Affiliations:** 1Laboratory of Molecular and Cellular Screening Processes, Center of Biotechnology of Sfax, University of Sfax, Sidi Mansour Road, P.O. Box 1177, Sfax 3018, Tunisia; 2CSIR-National Chemical Laboratory, Dr. Homi Bhabha Road, Pashan, Pune 411008, India; tejaschirmade@gmail.com (T.C.); ny.kadoo@ncl.res.in (N.K.); 3Academy of Scientific and Innovative Research (AcSIR), Ghaziabad 201002, India; 4Laboratory of Extremophile Plants, Centre of Biotechnology of Borj-Cédria, B.P. 901, Hammam Lif 2050, Tunisia; hananamohsen70@gmail.com; 5Field Crop Laboratory (LR16INRAT02), Institut National de la Recherche Agronomique de Tunisie (INRAT), University of Carthage, Tunis 1004, Tunisia; khalilkhamassi9@gmail.com; 6Department of Horticulture, Faculty of Agriculture, Ataturk University, Erzurum 25240, Turkey; sercisli@gmail.com; 7Division of Seed Science and Technology, ICAR-Indian Agricultural Research Institute, New Delhi 110012, India; ravianu1110@gmail.com; 8Unit of Biochemistry, Centre of Excellence for Biomaterials Engineering, Faculty of Medicine, AIMST University, Semeling, Bedong 08100, Malaysia

**Keywords:** oil palm, FAD2, oleic/linoleic acid ratio, fatty acids, in silico annotation, SNP, structural modeling

## Abstract

**Simple Summary:**

Palm oil has become the world’s most important vegetable oil in terms of production quantity, and its overall demand is exponentially growing with the global population. The fatty acid composition and particularly the oleic/linoleic acid ratio are major factors influencing palm oil quality. In this study, we focused on FAD2, a fatty acid desaturase enzyme involved in the desaturation and conversion of oleic acid to linoleic acid in *Elaeis oleifera*, identified through in silico annotation analysis. Our phylogenetic and comparative studies revealed two SNP markers, SNP278 and SNP851, significantly correlated with the oleic/linoleic acid contents. Our study provides fundamental insights into the mechanism of fatty acids synthesis in oil palm and could support the application of molecular biology techniques to enhance the enzymatic activity and substrate affinity of EoFAD2.

**Abstract:**

American oil palm (*Elaeis oleifera*) is an important source of dietary oil that could fulfill the increasing worldwide demand for cooking oil. Therefore, improving its production is crucial and could be realized through breeding and genetic engineering approaches aiming to obtain high-yielding varieties with improved oil content and quality. The fatty acid composition and particularly the oleic/linoleic acid ratio are major factors influencing oil quality. Our work focused on a fatty acid desaturase (FAD) enzyme involved in the desaturation and conversion of oleic acid to linoleic acid. Following the in silico identification and annotation of *Elaeis oleifera* FAD2, its molecular and structural features characterization was performed to better understand the mechanistic bases of its enzymatic activity. *EoFAD2* is 1173 nucleotides long and encodes a protein of 390 amino acids that shares similarities with other FADs. Interestingly, the phylogenetic study showed three distinguished groups where EoFAD2 clustered among monocotyledonous taxa. EoFAD2 is a membrane-bound protein with five transmembrane domains presumably located in the endoplasmic reticulum. The homodimer organization model of EoFAD2 enzyme and substrates and respective substrate-binding residues were predicted and described. Moreover, the comparison between 24 FAD2 sequences from different species generated two interesting single-nucleotide polymorphisms (SNPs) associated with the oleic/linoleic acid contents.

## 1. Introduction

In a healthy human diet, oil can have a major influence on an individual’s overall well-being. In particular, the types and the amounts of fatty acids consumed on a daily basis impact the long-term health of an individual [1,2,3]. In the last few years, palm oil has become the world’s most important vegetable oil in terms of production quantity, reaching almost 75 million metric tons in 2020 [4], and the overall demand for palm oil is exponentially growing with the increasing global population. Fatty acids are of increasing interest as they pertain to health issues and comprise the omega *ω*-6 and *ω*-3 polyunsaturated fatty acids [5,6]. Unsaturated fatty acids have one or more double bonds, with the (final) double bond occurring either six carbons (*ω*-6) or three carbons (*ω*-3) from the methyl (*ω*) end of the molecule. They may or may not contain functional groups on the same molecule [7]. Fatty acids in organisms, especially in plants, are the main structural constituents of organic membranes (phospholipid bilayers) and storage oils (neutral lipids) [8]. The fatty acid biosynthesis pathway is the primary metabolic pathway, since it is necessary for growth and development. The most important membrane phospholipids in all plant tissues are assembled using palmitate (C16:0) and oleate (C18:1) acyl groups [9].

Membrane glycerolipids have fatty acids connected to both the sn-1 and sn-2 positions of the glycerol backbone and a polar headgroup attached to the sn-3 position [10]. The mixture of nonpolar fatty acyl chains and a polar headgroup leads to the amphipathic physical properties of glycerolipids, which are indispensable for constructing membrane bilayers [11,12]. The fatty acid desaturases initiate the formation of a double bond into an acyl chain, involving the reduction of a molecule of O_2_ with the product being a fatty acid with a new double bond and water. At each double bond generated, the first hydrogen abstraction happens at the carbon atom closer to the carboxyl end. This is the rate-limiting step of desaturation [13]. The resulting double bond is naturally of cis conformation, although little quantities of trans conformation can happen [14]. The desaturation of fatty acids in the chloroplast and endoplasmic reticulum (ER) membrane complex lipids is also performed by the FADs (designated from FAD2 to FAD8) [15,16]. The FAD2 enzyme is an integral membrane phosphatidylcholine desaturase in the ER, acting on fatty acids at the sn-1 position [17,18]. The important function of FAD2 (also known as 18:1 desaturase) is to provide 18:2 and (following additional desaturation) 18:3, which are needed for the correct assembly of cellular membranes throughout the plant. Another essential role of this enzyme is synthesizing the polyunsaturated fatty acids in vegetable oils, which are the main source of essential fatty acids in humans [19,20,21]. The FADs are membrane proteins, supposed to include two iron atoms in their active site [22,23]. While several FAD2 cDNA configurations have been studied, only a few actual FAD2 gene sequences have been verified, the first being the single-copy *Arabidopsis* FAD2 gene [24].

*Elaeis oleifera* (Kunth) Cortés, also known as American oil palm, is an evergreen diploid tree species (2*n* = 32) native to South and Central America, and is commonly planted and used for edible oil production [25,26]. Improving palm oil as a source of *ω*-3 and *ω*-6 fatty acids for human health requires genetic knowledge of the palm, which producers can exploit. The biggest challenge is that the palm genome has not yet been fully annotated. Consequently, interest should be oriented to the genetic mechanisms underlying *ω*-3 and *ω*-6 fatty acid biosynthesis in the palm [27,28]. Therefore, this study aimed to characterize at the molecular and structural levels the putative genes responsible for encoding the fatty acid desaturase that most likely confers catalytic mechanisms in palm *Elaeis oleifera*, Δ12 fatty acid desaturase (EoFAD2).

Gene annotation is the identification, characterization, and localization of the functional genetic elements present within the genes, notably the coding regions, to allow the prediction of certain protein functions. There are two gene annotation methods, extrinsic and intrinsic. The intrinsic approach involves the evaluation of certain properties of the gene sequence without explicit comparison with other sequences. The relevant properties may include the ORF length, the presence or absence of ribosome-binding sites at the suitable distance upstream of the initiation codon, codon usage, and various statistical parameters that have been found to be typical of known genes. Conversely, the extrinsic approach includes the comparison of the putative amino acid sequence of the gene with protein sequence databases, as well as identifying functional motifs in the gene sequence using database searches. If the translated amino acid sequence of a putative ORF exhibits significant sequence homology to one or more proteins in the database, it is almost certain that the ORF in question is a real gene [29].

## 2. Materials and Methods

### 2.1. FAD2 Sequence Identification, Annotation, and Collection

Both the extrinsic and intrinsic methods of gene annotation were used and combined in our study. Among the extrinsic search methods that allow gene annotation, the “BLAST” search method (http://blast.ncbi.nlm.nih.gov/Blast.cgi) (accessed on 12 May 2021) was applied to find the partial *fad2* gene sequence of *Elaeis oleifera* as a query to obtain significant hits of similar sequences from other species. We used the BLASTX and TBLASTN programs to search the partial nucleic acid sequence of EoFAD2 with the nr protein database. Then, we applied the intrinsic method to predict coding structures using the DNA sequence that translates into protein. To achieve this, we used the program “MEME” (Multiple Em for Motif Elicitation; version 4.9, http://meme-suite.org/tools/meme, accessed on 12 May 2021), which discovers novel, ungapped motifs or conserved regions in target sequences. It identifies functional sites in a protein, which could be an active site, an interaction-binding site, etc. [30]. The final step of this in silico annotation was to manually verify and correct this annotation to justify the obtained results. Twenty-four gene sequences of FAD2 belonging to different oilseed species were collected from NCBI (http://www.ncbi.nih.gov, accessed on 12 May 2021) (Table 1). Eight oilseed crops were considered, namely American oil palm (*Elaeis oleifera*), soybean (*Glycine max*), colza (*Brassica napus*), sunflower (*Helianthus annuus*), peanut (*Arachis hypogaea* and *Arachis monticola*), corn (*Zea m**ays*), and sesame (*Sesamum indicum*).

### 2.2. Physico-Chemical Properties of the FAD2 Protein

The protein sequences of *fad2* genes were analyzed using several bioinformatics tools. Comparative and bioinformatics analyses of FAD2 protein sequences were performed online on the NCBI (http://www.ncbi.nih.gov, accessed on 12 May 2021) and EXPASY (http://expasy.org/tools, accessed on 12 May 2021) platforms. The MEME webserver (http://meme-suite.org/tools/meme, accessed on 12 May 2021) was used to find conserved domains or sequence patterns often associated with a biological role [30]. Selected parameters of the MEME search were as follows: minimum width for each motif, 6; maximum width for each motif, 50; maximum number of motifs to find, 10; and occurrence of each motif, 0 or 1 per sequence. A multiple sequence alignment of the downloaded FAD2 protein sequences was performed using ClustalW with default parameters. A neighbor-joining phylogenetic tree was constructed using MEGA version 11 (Molecular Evolutionary Genetics Analysis; https://www.megasoftware.net/, accessed on 12 May 2021) based on aligned sequences of FAD2 proteins [31].

### 2.3. Bioinformatics Analysis, Three-Dimensional (3D) Protein Structure Modeling, and Docking Analysis

The 3D structure of the EoFAD2 protein was predicted *de novo*, as there were no experimentally determined structures for FAD2 homologs in the PDB database. The I-TASSER (Iterative TASSER) webserver tool (https://zhanggroup.org/I-TASSER/, accessed on 12 May 2021) was used to predict the structure of EoFAD2 de novo [32]. Multiple threading alignments of different PDB templates were performed using LOMETS2 to model the protein in parts based on structural conservation [33]. Transmembrane domain prediction was performed using TMHMM [34] and TMPred [35]. Protein–protein interactions to form a homodimer were predicted using ClusPro2.0 [36]. The PyMol Molecular Graphics System v1.7.4.4 Edu (https://pymol.org/support-platforms.html, accessed on 12 May 2021) was used to visualize the protein structures. Local quality assessment was performed using QMEANBrane in the SWISS-MODEL workspace [37].

Molecular protein docking of oleoyl glycerol as the ligand and the predicted structure of EoFAD2 from *Elaeis oleifera* as the receptor was performed using AutoDock Vina v1.2 [38]. The amino acid residues predicted to interact with the fatty acyl glycerol were incorporated in the grid box. The conformation with the highest predicted affinity was considered further. FAD2 protein sequences from other species were also analyzed for amino acid substitutions predicted to be involved in the substrate binding. Molecular docking was performed using the substituted amino acid.

Molecular modeling and docking analyses were combined to study the structure–function relationship of the poorly structurally characterized integral membranes of FAD2. The findings allowed us to investigate the role of each substitution within the *fad2* gene sequences of the studied oilseeds species.

### 2.4. Interspecific FAD2 SNPs Identification

Multiple sequence alignment was performed using ClustalW (https://www.genome.jp/tools-bin/clustalw, accessed on 12 May 2021) [39] to investigate the presence of polymorphisms, including single nucleotide polymorphisms (SNPs), among the FAD2 sequences of different oilseed species.

### 2.5. Statistical Analysis

The correlation confirmation between two predicted interspecific SNPs and the synthesis of the two fatty acids, oleic (C18:1) and linoleic (C18:2), was performed using Fisher’s exact test. This test is used for the comparison of proportions in randomized trials. Fisher’s exact test can be particularly conservative, especially when the sample sizes are small or when the observed ratios are close to 0. The test was carried out using the exact2x2 R package (https://cran.r-project.org/, accessed on 12 May 2021).

## 3. Results and Discussion

### 3.1. EoFAD2 Identification and Annotation

EoFAD2 was identified and annotated following the described procedure, as indicated above in Section 2. The EoFAD2 mRNA sequence was 1173 nucleotides long and encoded a protein of 390 amino acids that shared similarities with other FADs. Both nucleotide and peptide sequences were submitted to the Genbank database under the accession numbers KY006847.1 and APO15326.1, respectively.

Several biological roles and functions have been attributed to FAD2, namely their key role in fatty acids biosynthesis (mainly linoleic acid), plant development (vegetative growth, germination, and anther development), and abiotic stress tolerance (salinity, drought, and cold) as mentioned for several species in Table 2.

### 3.2. Comparative Analysis and Phylogenetic Study of fad2 Genes from Several Oilseed Species

The study included 24 *fad2* gene sequences from eight oilseed species: American oil palm, soybean, canola, sunflower, peanut (cultivated and wild relative), maize, and sesame (Table 1). The multiple sequence alignment of the FAD2 sequences performed using ClustalW made the drawing of the phylogenetic tree possible. Indeed, the phylogenetic analysis revealed the differences and similarities in structural organization among the different sequences. The tree generated by the Neighbor-Joining (NJ) distance method highlighted three distinct groups (Figure 1):(a)The first group contained FAD2 members of six species: *Glycine max*, *Arachis hypogaea*, *Arachis monticola*, *Helianthus annuum*, and *Sesamum indicum*;(b)The second group corresponded to a single species: *Brassica napus*;(c)The third group included *Zea mays* with *Elaeis oleifera*.

In addition, the conserved motifs of FAD2 proteins were obtained by MEME analysis, and are represented in different colored boxes that are described in Figure 1. Ten motifs (named motif 1 to motif 10) were detected (Figure 1). Particularly, *H. annuus*, *S. indicum*, *Z. mays*, and *E. oleifera* lack motifs one and two in comparison with the other species.

The ZmFAD2 sequence corresponded to a cDNA of the *fad2* gene of *Zea mays*, and the predicted sequence “KY006847.1” was well separated from the other groups. This could be explained by the membership of *Zea mays* and *Elaeis oleifera* to the monocotyledonous class, unlike the other species that were dicotyledonous. The first and the second groups of dicotyledonous plants, which represented the majority of studied accessions, could be divided into three major subgroups: A, B, C, which mainly included Leguminosae (*Glycine max* and *Arachis*), Pedaliaceae (*Sesamum indicum*), and Asteraceae (*Helianthus annuus*), respectively (Figure 1A). From this phylogenetic analysis, it can be deduced that the study of the genetic relationship and the molecular evolution of species could be exploited to understand the genes implicating in metabolic pathways, such as the genes involved in the metabolism of fatty acids. Therefore, we carried out a study on the *fad2* gene of *Elaeis oleifera*, responsible for the transformation of the monounsaturated fatty acid (C18:1) into polyunsaturated fatty acid (C18:2), to understand the relationship between the acidic composition of the various vegetable oils and their nucleotide and protein sequences of the *fad2* gene.

### 3.3. Estimate of Evolutionary Divergence among the Sequences

We carried out a progressive alignment of the iterative construction of the multiple alignments by grouping the pairwise alignments of the *fad2* gene sequences of the studied species. This analysis was performed through three main steps: the first step consisted of the alignment of all possible pairs and establishing a distance matrix based on alignments scores (Table 3). The second step was the construction of a guide shaft using the previously calculated alignment distances. The third step was to realign the sequences or groups of sequences in the order determined by the guide shaft. The mean pairwise distance value between the aligned sequences was 0.30, with a very small value (near 0) between the sequences belonging to the same species and a maximum value of 0.57 between the sequences of the species *Glycine max* and the sequence of the species *Zea mays*. This was probably due to the fact that the two species are distantly related. Indeed, maize is a monocot, whereas soybean is a dicot. However, this similarity analysis revealed that the *fad2* gene is conserved across taxa, as it is involved in fatty acid synthesis, which is an essential pathway.

In addition, we note from Table 3 that the two π and θw values were slightly distant. This indicates a low nucleotide diversity (π = 0.26) in the coding region of the *fad2* gene of the predicted sequence and the sequence from *Zea mays*. This value was higher than observed values for other plants, such as *Arabidopsis thaliana* (π = 0.0079) [79]. From these results, we can suggest that the level of nucleotide polymorphism was strictly affected by the genetic origin of the population on the one hand and the number and nature of the selected gene on the other hand. In addition, we noticed that the value of π = 0.26 was higher than that of θw (0.16), indicating no significant difference, and, subsequently, the polymorphism of the coding region of the *fad2* gene fitted into the neutral mutation hypothesis. In addition, the non-significant value of the D statistic (2.57) of the Tajima test (*p* > 0.05) reported for this gene also tended to confirm this hypothesis.

### 3.4. Comparative Analysis between Interspecific FAD2 SNPs

#### 3.4.1. In Silico SNP Prediction

The annotation of the studied sequences allowed us to discover two SNPs considered interesting in the involvement of the *fad2* gene for the conversion of oleic acid to linoleic acid. The first SNP373 was a transition (conversion from T to C). The ‘T’ variant was detected in the nucleotide sequences of two species (sunflower and sesame), while the other variant with ‘C’ was detected in three other species, namely corn, peanut, and American oil palm. The change of T to C nucleotides in the nucleotide sequences of the *fad2* gene of *Elaeis oleifera* resulted in the substitution of the hydrophilic amid amino acid Q (Glutamine) to the hydrophilic hydroxyl amino acid S (Serine). The second SNP718 was also a transition (conversion from A to G). The ‘G’ variant was detected in the nucleotide sequences of three species (American oil palm, peanut, and corn). In comparison, the other ‘A’ variant was detected in the sunflower and sesame species. The change of the A to G nucleotide in the *fad2* gene did not result in a change of the serine amino acids. An intraspecific FAD2 polymorphism study in *Olea europaea* identified an interesting correlation between some SNPs and the oleic/linoleic acid ratio [80].

#### 3.4.2. Genotype/Phenotype Association Study of Predicted SNPs

To confirm the correlation between these two SNPs and the synthesis of the oleic (C18:1) and linoleic (C18:2) acids, we performed statistical analyses using Fisher’s exact test. This test is used for the comparison of proportions in randomized trials. Fisher’s exact test can be particularly conservative, especially when the sample sizes are small or when the observed ratios are close to 0. Thus, we were able to show two highly significant associations of the first SNP373 with the accumulation of the two unsaturated fatty acids, oleic acid (*p* = 0.006) and linoleic acid (*p* = 0.013). Indeed, the contents of monounsaturated fatty acids (C18:1) were significantly high (average of 54.10 ± 11.95) in species carrying the homozygous CC genotype at SNP373, which included peanut. The content of monounsaturated fatty acids was also high in sesame, corn, and American oil palm, which had the heterozygous TC genotype. On the other hand, the level of polyunsaturated fatty acids (C18:2) was significantly high (average of 57.43 ± 5.75) in species carrying the homozygous TT genotype at SNP373, which included sunflower, as well as the heterozygous TC genotype found in sesame and corn. Indeed, we can suggest that peanuts carrying the CC homozygous allele would synthesize the most monounsaturated fatty acid (C18:1) compared to the TT homozygous species (sunflower) and the TC heterozygous species (sesame, corn, and American oil palm).

Similarly, the species with the TT genotype at the SNP373 marker accumulated less linoleic fatty acid (C18:2) than the other CC and TC species (Table 4). Moreover, we were able to derive two highly significant associations of the second SNP718 with the accumulation of the two unsaturated fatty acids: oleic acid (C18:1; *p* = 0.007) and linoleic acid (C18:2; *p* = 0.002). Indeed, the monounsaturated fatty acid content (C18:1) was significantly high in species carrying the homozygous GG genotype, with a mean of 58.70 ± 9.34, which included peanut (*Arachis hypogaea* and *Arachis monticola*), while the polyunsaturated fatty acid content (C18:2) was significantly high in species carrying the homozygous AA and AG genotypes, with a mean of 53.75 ± 8.74, which included sunflower, corn, and sesame (Table 4). Both these markers are located within the *fad2* gene, which is involved in the synthesis of polyunsaturated fatty acids and, more specifically, in the transformation of oleic acid (C18:1) to linoleic acid (C18:2) (Vance et al., 2001). Thus, this suggests a direct effect of the allelic variation of the markers SNP373 and SNP718 on the fatty acid composition, namely oleic acid and linoleic acid, for each species studied. For the remaining saturated and unsaturated fatty acids, the comparison of means tests between species showed no significant association with SNP373 or SNP718. This suggests the hypothesis that these two polymorphisms are substitutions that are directly involved in protein functions, which explains the variation in the contents of C18:1 and C18:2 (Table 4).

### 3.5. EoFAD2 Protein Features, 3D Structure Prediction, and Docking Analysis

#### 3.5.1. Structure Prediction

FAD2 is a membrane-bound protein found in the endoplasmic reticulum of plant cells. EoFAD2 has a size of 44.1 kD and a pI of 8.4 (Table 5). The protein is predicted to be organized in an α-helix structure (28%), extended strand (21%), and has 51% random coil structure (Table 5). As shown in Table 6, leucine, alanine, valine, and proline are the most abundant aa in EoFAD2. The 3D structure of this enzyme was predicted de novo using the I-TASSER server, since there is no structure homologous to the FAD2 present in the Protein Data Bank (PDB). The I-TASSER tool usually predicts more than one structure for the query sequence, and, in this study, five models were predicted (Appendix A). Of the five, the best model (Figure 2A,C) was selected based on the C-score, positions of the transmembrane domains (Figure 2B), and structure assessment using PROCHECK. Energy minimization using OPLS-AA was performed on the initial structure obtained from I-TASSER to correct any bond clashes and refine the stereochemistry of the predicted protein (Appendix A). The PROCHECK tool was used to assess the stereological quality of the structure. The Ramachandran plot for the structure showed 91.2% of the total residues (including glycine and proline) located in the core and additionally allowed regions, with 6.3% of the residues in the generously allowed region, and only 2.5% were marked as outliers.

FAD2 enzymes in the endoplasmic reticulum membrane usually form heterodimers and homodimers for their function [81]. We further predicted the homodimer structure of EoFAD2 using the ClusPro2.0 tool, considering the hydrophobic interactions (Figure 3A). TMPred and TMHMM tools were used to predict the transmembrane domains present in the protein. Both tools provided similar results indicating overlapping regions as transmembrane domains. Five transmembrane domains were identified with about 19–23 residues in each transmembrane domain, and the results are summarized in Table 7. The estimation of the local quality of the predicted membrane protein was evaluated using QMEANBrane, and the membrane insertion energy was found within the limits (Appendix A). I-TASSER also provided preliminary predictions of the possible substrates and respective substrate-binding residues. The residues 111, 117, 149, 152, 153, 281, 322, 325, and 326 were predicted to be involved in metal ion binding. FAD2 enzymes are metal ion-binding proteins and contain ferrous ions. However, the total number of metal ions in a monomer is unknown. The residues 114, 118, 120, 121,122, 127, and 142 were predicted to be for oleoyl–glycerol-binding (Figure 3B). Moreover, Figure 3C illustrates the substrate oleoyl glycerol docked with the predicted structure of *Elaeis oleifera* FAD2. In fact, the amino acid residues represented in the figure are the predicted interacting residues with the substrate, meaning that the residues predicted in binding with oleoyl glycerol were considered to predict the binding pocket during the molecular docking studies. The best binding affinity was predicted to be −5.9 kcal/mol for the conformation represented in Figure 3C.

#### 3.5.2. Docking Analysis between Oilseed Species

The sequence alignment of the five species—sunflower, corn, sesame, peanut, and American oil palm—showed numerous substitutions, but, in structural analysis, only nine could be studied. The amino acids at positions 114, 118, 120, 121, 122, 127, 142, and 240 were conserved throughout the different species. The only notable change was at position S124Q. The docking results are shown in Figure 4 with the change in the amino acid. In fact, the differences in substrate binding may be influenced by amino acids other than the substrate-binding pocket.

FAD2 sequences from other species (Figure 4A,B) were also analyzed to identify any changes in the residues predicted for interacting with the substrate. However, the residues were conserved throughout the sequences, and any change in the substrate specificity or affinity may be attributed to the differences in the amino acid sequence other than the binding pocket.

## 4. Conclusions

The identification and characterization of EoFAD2, a fatty acid desaturase enzyme involved in the desaturation and conversion of oleic acid to linoleic acid, is the first and a preliminary step for American oil palm improvement. The phylogenetic and comparative studies showed similarities and variability among the *fad2* gene sequences obtained from eight oilseed species. Interestingly, two SNP markers, SNP373 and SNP718, associated with the oleic/linoleic acid contents were identified. The homodimer structures of EoFAD2 and oleoyl–glycerol substrate-binding residues were predicted and described. Docking analysis was used to study structure-function relationship of the poorly structurally characterized integral membrane protein FAD2. The findings allowed us to investigate the role of the substitution S124Q among the FAD2 sequences of the studied oilseed species. The results suggested the fundamental role of this genetic substitution in reducing or enhancing the FAD2 activity to obtain more or less oleic acid content, respectively.

Altogether, these results enhanced our understanding of the mechanism of fatty acids synthesis in American palm and could facilitate the use of molecular biology techniques to improve the enzymatic activity and substrate affinity of EoFAD2. 

## Figures and Tables

**Figure 1 biology-11-00529-f001:**
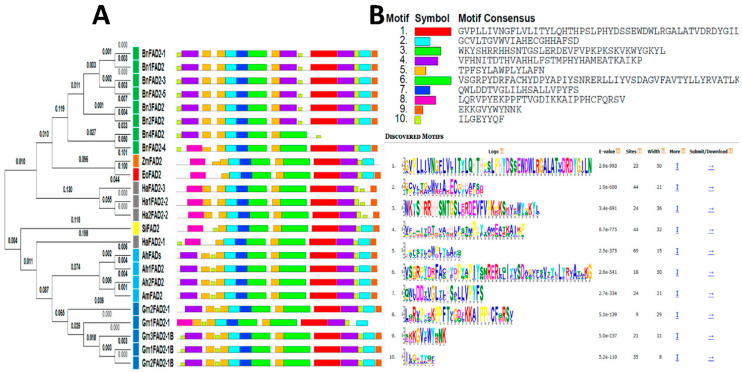
(**A**) Phylogenetic tree of 24 FAD2 protein sequences from eight oilseed species studied using the MEGA program and neighbor-joining analysis with the schematic representation of their conserved motifs identified by the MEME server, described in (**B**).

**Figure 2 biology-11-00529-f002:**
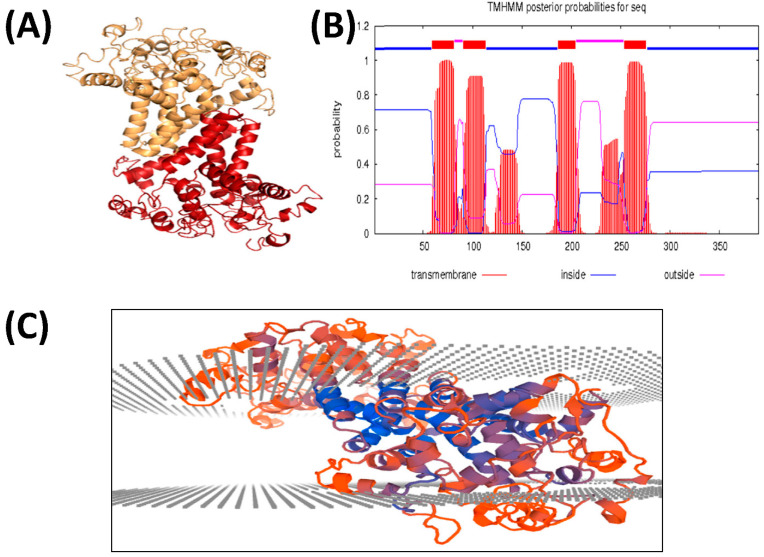
(**A**,**C**) 3D predicted model of the EoFAD2 protein; (**B**) Predicted transmembrane domains in FAD2 by TMHMM server (CBS; Denmark).

**Figure 3 biology-11-00529-f003:**
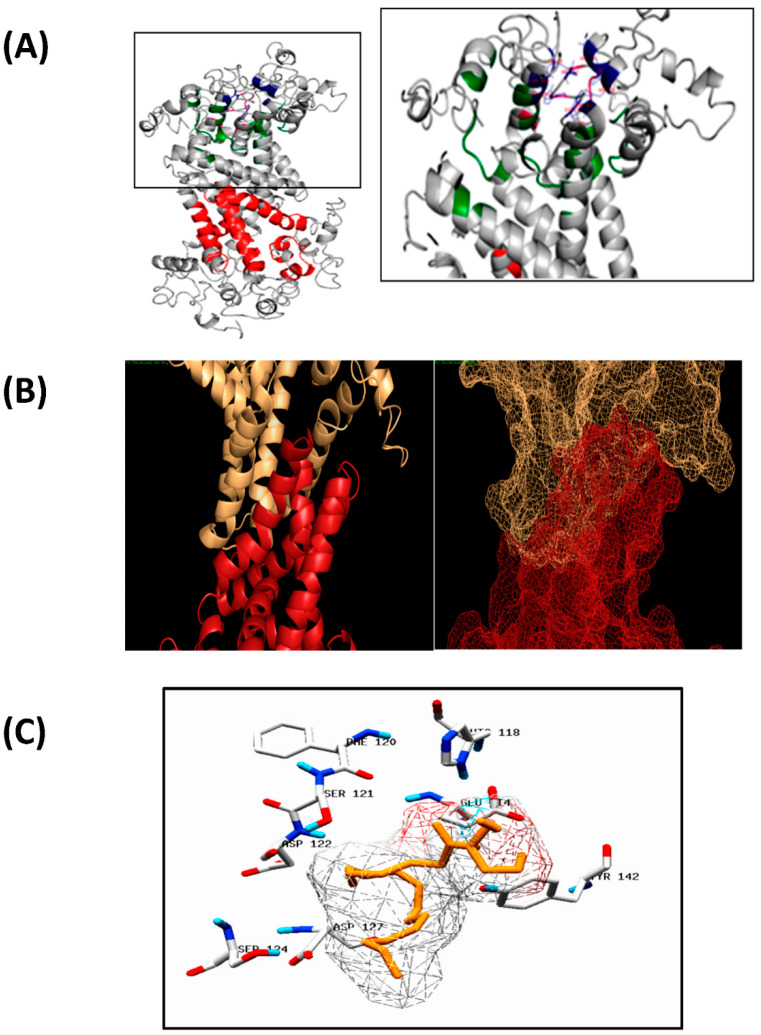
(**A**) EoFAD2 homodimer organization taking into account the hydrophobic interactions; (**B**) Potential substrates and substrate-binding residues; (**C**) Substrate oleoyl glycerol docked with the predicted structure of *Elaeis oleifera* FAD2.

**Figure 4 biology-11-00529-f004:**
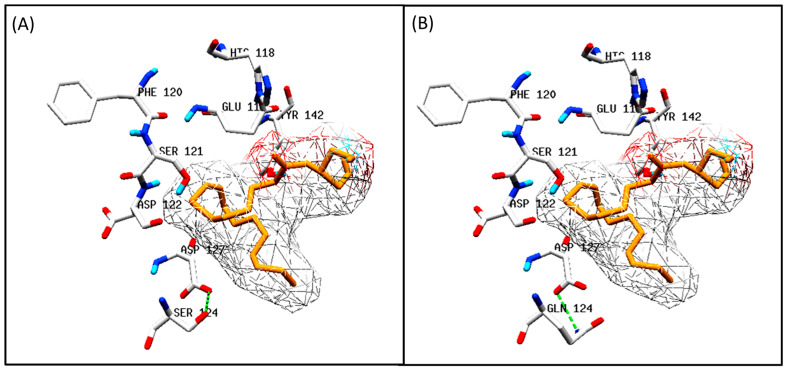
Molecular docking of oleoyl glycerol as the ligand and the predicted structure of EoFAD2 from *Elaeis olifera* as the receptor (**A**) and the other studied oilseed species (**B**) performed using AutoDock Vina v1.2.

**Table 1 biology-11-00529-t001:** GenBank accession numbers of 24 *FAD2* gene sequences from eight oilseeds species.

Species	Accession Number	Code
*Elaeis oleifera*	KY006847.1	EoFAD2
*Glycine max*	AY611472.1	Gm1FAD2-1
	L43920.1	Gm2FAD2-1
	EU908061.1	Gm1FAD2-1B
	EU908062.1	Gm2FAD2-1B
	DQ532370.1	Gm3FAD2-1B
*Brassica napus*	FJ907397.1	BnFAD2-1
	FJ907399.1	BnFAD2-3
	FJ907401.1	BnFAD2-4
	FJ907400.1	BnFAD2-5
	AF243045.1	Bn1FAD2
	DQ767949.1	Bn2FAD2
	AY577313.1	Bn3FAD2
	AY592975.1	Bn4FAD2
*Helianthus annuus*	AY800245.1	HaFAD2-1
	AY803008.1	HaFAD2-3
	AY802997.1	Ha1FAD2-2
	AY802995.1	Ha2FAD2-2
*Arachis hypogaea*	AF030319.1	AhFADs
	DQ019933.1	Ah1FAD2
	AF248739.1	Ah2FAD2
*Arachis monticola*	AY900663.1	AmFAD2
*Zea m* *ays*	AB257309.1	ZmFAD2
*Sesamum indicum*	AF192486.1	SiFAD2

**Table 2 biology-11-00529-t002:** Different functions and biological roles attributed to FAD2 enzymes within several species through biological engineering and polymorphism studies.

Gene ID/Name	Species	Function	Reference
L26296	*Arabidopsis thaliana*	Polyunsaturated lipid synthesis, salt tolerance during seed germination and early seedling growth, and vegetative growth	[24,40,41]
AY733076, AY733077	*Olea europaea*	Linoleic acid synthesis, wounding response of olive fruit mesocarp, cuticle formation, and cold-acclimation, and abiotic stresses (drought/cold) response	[42,43,44,45,46]
AB094415	*Spinacia oleracea*	Linoleic acid synthesis	[47]
CtFAD2-1-11	*Carthamus tinctorius*	Linoleic acid synthesis and crepenynic acid synthesis (with C16:1 as a substrate)	[48]
AsFAD2-1-24	*Artemisia* *sphaerocephala*	Linoleic and palmitolinoleic acid biosynthesis	[49]
EF186911	*Arachis hypogaea*	Enhancement of peanut oil quality	[50,51]
MF 693460	*Idesia polycarpa*	Linoleic acid accumulation	[52]
GmFAD2–1a GmFAD2–1b	*Glycine max*	Increased oleic acid content in mutant lines	[53,54]
JX964741, JX964747	*Crambe abyssinica*	Polyunsaturated fatty acids biosynthesis	[55]
HaFAD2-1-11	*Helianthus annuus*	Linoleic acid synthesis and storage oil desaturation in seed	[56]
AAX11454, ACP39503, ACF49507, ABK59093, AAS19533, AEI60129, ACZ06072, AAN87573, AAF04094	*Sesamum indicum*, *Brassica napus*, *Linum usitatissinum*, *Ricinus communis*, *Cucurbita pepo*, *Vitis labrusca*, *Arachis hypogaea*, *Vernicia fordii*, *Vernonia galamensis*	Linoleic acid synthesis and storage oil desaturation in seed	[57]
OsFAD2-1 RNAi	*Orysa sativa*	Alteration of lysophospholipid composition in the endosperm of rice grain and influence on starch properties	[58]
AF331163	*Gossypium hirsutum*	Linoleic acid accumulation, anther development, and cold and light responsiveness	[59,60,61]
MF318524	*Bidens pilosa*	Polyacetylene biosynthesis	[62]
XM_019004668.1, XM_019004667.1, XM_018993369.1, XM_018993367.1	*Juglans regia*	Linoleic acid synthesis	[63]
AF243045	*Brassica napus*	Promotion of seed germination and hypocotyl elongation	[64]
DQ31678	*Populus tomentosa*	Freezing tolerance	[65]
SalFAD2.LIA1, SalFAD2.LIA2	*Sinapis alba*	Linoleic acid accumulation	[66]
KT023602	*Elaeis guineensis*	Linoleic acid synthesis	[67]
Partial RtFAD2	*Reutealis trisperma*	Regulation of fatty acid desaturation	[68]
Csa3M808360, Csa4M286360	*Cucumis sativus*	Temperature stress responsiveness	[19]
FAD2 (SNPs)	*Cucurbita moschata*, *Cucurbita maxima*, *Cucurbita pepo*, and *Cucurbita ficifolia*	Linoleic acid synthesis	[69]
CsFAD2 (SNPs)	*Camelina sativa*	Linoleic acid synthesis and storage oil desaturation in seed	[70,71]
EU275211	*Davidia involucrata*	Linoleic acid synthesis	[72]
X91139, EF639848, FJ696650, FJ696651, FJ696652	*Brassica juncea*	Biosynthesis of polyunsaturated fatty acids in seeds and cold responsiveness	[73,74]
DQ496227, ZmFAD2 (SNPs)	*Zea mays*	Ratio of oleic/linoleic acid	[75,76]
GU353167	*Jatropha curcas*	Conversion of oleic acid to linoleic acid in the seed	[77]
HQ171179, HQ171180	*Phaseolus lunatus*	Cold, drought, and salt stress responsiveness	[78]

**Table 3 biology-11-00529-t003:** Overview of the relationship between all studied sequences of the *fad2* gene; (detect deviations from the neutral model).

Sequences	Region	*n*	s	π	D	θw
All Sequences	1–1172	24	0.60	0.26	2.57	0.16

*n* = number of sequences; s: substitution, π: the level of nucleotide diversity over the entire sequence; θw: the level of nucleotide diversity per site; D: the D statistic of the Tajima test.

**Table 4 biology-11-00529-t004:** Association between predicted SNPs (SNP373 and SNP718) and the composition of fatty acids for the studied vegetable oils.

	C18:0		C18:1		C18:2		C18:3	
	Mean ± SD	*p*-Value	Mean ± SD	*p*-Value	Mean ± SD	*p*-Value	Mean ± SD	*p*-Value
**SNP373**								
C	2.972 ± 1.154	*0.754*	54.100 ± 11.948	* **0.006** *	23.747 ± 14.195	* **0.013** *	2.6150 ± 4.674	*0.902*
T	3.283 ± 1.338		21.333 ± 3.666		57.433 ± 5.749		3.053 ± 4.030	
**SNP718**								
G	2.530 ± 0.907	*0.282*	58.700 ± 9.337	* **0.007** *	17.430 ± 7.925	* **0.002** *	3.486 ± 5.311	*0.735*
A	3.537 ± 1.205		26.075 ± 9.944		53.750 ± 8.735		2.29 ± 3.627	

*p*: *p*-value Fisher’s test; SD: Standard deviation. Italic format: the *p*-value; Italic and bold formats: *p*-value < 0.05.

**Table 5 biology-11-00529-t005:** Peptidic composition and physico-chemical features of the EoFAD2 protein.

Properties	Length (aa)	MW (kD)	pI	TM Regions	α-Helix Structure (%)	Extended Strand Structure (%)	Random Coil Structure (%)
EoFAD2	390	44.1	8.4	5	28	21	51

**Table 6 biology-11-00529-t006:** Most frequent aa in the EoFAD2 protein.

Aa (%)	Leu	Ala	Val	Pro	Ser	Arg	Gly	Tyr	His
EoFAD2	10	9.7	7.7	7.4	6.4	6.2	6.2	5.4	5.1

**Table 7 biology-11-00529-t007:** Transmembrane domains as predicted by TMPred.

TM Position	Length	Orientation
61–85	19	o-i
94–114	21	i-o
128–148	21	o-i
190–209	20	i-o
257–279	23	o-i

i = inside, o = outside.

## Data Availability

All new research data were presented in this contribution.

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
