# Peer review of "Comparative Analysis and Structural Modeling of Elaeis oleifera FAD2, a Fatty Acid Desaturase Involved in Unsaturated Fatty Acid Composition of American Oil Palm"

_biology, 2022, doi:10.3390/biology11040529_

Round 1

Reviewer 1 Report

This manuscript focuses on the in silico characterization of the fatty acid desaturase FAD2 protein from Elaeis oleifera. Protein motifs were analyzed, and 3D structure was predicted. While the manuscript is globally well organized, I think that several points need to be addressed to improve the quality of the manuscript. Because the results do not show any functional characterization of FAD2 and its real impact on fatty acid composition, the title “Elaeis oleifera FAD2, a fatty acid desaturase controlling oleic/linoleic acid ratio in American oil palm” is not accurate and should be changed, I think. Moreover, conclusions and discussion of the results could be extended, and more references could be added to support the discussion. Please find below my detailed suggestions.

  • A functional validation (at least in vitro) of the predicted activity of EoFAD2 protein would have greatly improved the significance of this study.
  • Line 105: “the described procedure ». Where is it described? A reference could be added here.
  • Figure 1:
    • distances could be indicated along the branches of the tree.
    • The color code should be explained in the legend. The motif numbers should be indicated.
    • The content of the supplementary Figure 1 could be added to the figure 1 (panel A would show the tree, panel B would present the motif sequences). I think it would be easier for the reader to have all the information concerning the domains in a same figure.
    • Line 182: “PDB” abbreviation should be explained.
  • Figure 3:
    • Would it be possible to represent the substrate within the substrate binding pocket?
    • Figure 3B: I do not see potential substrates. It would be more informative to visualize the protein with the potential substrates.
  • Paragraph 2.4.2: references should support the results and the discussion of the results. FAD2 proteins from other species that were functionally/biochemically characterized should be mentioned, especially FAD2 proteins of the species for which SNP were identified.
  • Supplemental Figure 2: the labels of the models 1, 2 and 3 are not visible on my screen. Labels should appear clearly on the final pdf version.
  • Line 64: “diminution” should be replaced by “reduction”, I think.

Author Response

Dear Reviewer,

Please see the attachment point by point response of the comments of the Reviewer 1

Reviewer 2 Report

In this work, the authors identified a gene sequence putatively encoding a fatty acid desaturase FAD2 in American oil palm.  Sequence analysis was performed for this gene and its deduced protein, a desaturase enzyme. Two SNP markers, SNP278 and SNP851, possibly associated with oleic/linoleic acid contents were identified. Results from sequence analysis of Elaeis oleifera FAD2 aligned with other homologous sequences from several oily species are predictable and provide fundamental information for further investigation. The main scientific merit in this manuscript is the identification of two SNP markers, SNP278 and SNP851, which may be related to or responsible for the ratio of oleic/linoleic acid in seed oils.

However, the authors did not provide any convincing evidence to prove this observation. It is also possible that the two SNP markers might be incidentally observed with the preferential oleic/linoleic acid conversion of the desaturase. Thus, the authors are suggested to provide scientific data to further demonstrate how SNP278 and/or SNP851 are responsible for the effect on desaturase activity for the conversion of oleic acid to linoleic acid. The authors may provide experimental data to show that a single mutation of FAD2 at 278 or 851 residue comparably causing the change of the desaturase activity. Alternatively, they may convincingly show detailed structure-function relationship via molecular modeling to indicate that the single mutation induces a conformational shift of the enzyme active site and results in the preferential oleic/linoleic acid conversion of the desaturase theoretically.   

Author Response

Dear Reviewer,

Please see the attachment point by point the responses of the reviewer's comments.

Best regards

Round 2

Reviewer 2 Report

The authors adequately answered my major concern in the revised manuscript. Thus, this manuscript is recommended for publication.